# Zinc Laurate Protects against Intestinal Barrier Dysfunction and Inflammation Induced by ETEC in a Mice Model

**DOI:** 10.3390/nu15010054

**Published:** 2022-12-22

**Authors:** Qianqian Chen, Peng Wang, Jinrong Wang, Jilong Xu, Cen Liu, Hanzhen Qiao, Liping Gan, Erzhen Duan, Yihui Zhang, Meiying Wang, Xujing Wu, Xinyu Du, Liying Li

**Affiliations:** College of Biology Engineering, Henan University of Technology, Zhengzhou 450001, China

**Keywords:** zinc laurate, enterotoxigenic *Escherichia coli*, intestinal inflammation, intestinal barrier, virulence factors

## Abstract

Enterotoxigenic *Escherichia coli* (ETEC) infection is one of the most common bacterial causes of diarrhea in children and young farm animals. Medium-chain fatty acids (MCFAs) have been widely used for their antibacterial and immune functions. However, there is limited information regarding the role of MCFAs chelated with Zn in diarrhea induced by ETEC infection. Here, zinc laurate (ZnLa) was used to evaluate its protective effect in a mice diarrhea model induced by ETEC. A total of 45 ICR-weaned female mice were randomly assigned to marginal zinc deficiency (dZn), dZn, and ETEC infection groups (dZn+ETEC); ETEC infection was co-treated with a low, middle, or high dose of ZnLa (ZnLa LOW+ETEC, ZnLa MID+ETEC, and ZnLa HIGH+ETEC), respectively, to explore the effect and its mechanism of ZnLa on diarrhea and intestinal health of mice challenged with ETEC. To further compare the antibacterial efficiency of ZnLa and ZnSO_4_ in mice with ETEC infection, a total of 36 ICR-weaned female mice were randomly divided into ZnLa, ZnLa+ETEC, ZnSO_4_, and ZnSO_4_ and ETEC infection groups (ZnSO_4_+ETEC); moreover, the growth curve of ETEC also compared ZnLa and ZnSO_4_ in vitro. Mice pretreated with ZnLa were effectively guarded against body weight losses and increases in diarrhea scores induced by ETEC. ZnLa pretreatment also prevented intestinal barrier damage and ion transport in mice challenged with ETEC, as evidenced by the fact that the intestinal villus height and the ratio of villus height and crypt depth, tight junction protein, and Na^+^ absorption were higher, whereas intestinal permeability and anion secretion were lower in mice pretreated with ZnLa. In addition, ZnLa conferred effective protection against ETEC-induced intestinal inflammatory responses, as the increases in protein and mRNAs of proinflammatory cytokines were prevented in serum and jejunum, which was likely associated with the TLR4/MYD88/NF-κB signaling pathway. The increase in ETEC shedding and virulence-related gene expression was prevented in mice with ZnLa pretreatment. Finally, the growth of ETEC and virulence-related gene expression were lower in the ZnLa group than in ZnSO_4_ with an equal concentration of zinc. These findings suggest that ZnLa is a promising prevention strategy to remedy ETEC infection.

## 1. Introduction

Enterotoxigenic *Escherichia coli* (ETEC) is the most common pathogen of bacillary dysentery, causing a high incidence of intestinal diseases and diarrhea in newborns and young farm animals [1]. The colonization of ETEC in the intestinal epithelium and the secretion of enterotoxins are two key processes that initiate infection. Firstly, ETEC expresses one or more membrane adhesins, which help it colonize the intestinal mucosa surface and bind closely to glycoprotein receptors on the brush margin of intestinal cells. After colonization, the enterotoxins, including heat-labile enterotoxins (LT) and heat-stable enterotoxins (ST), are secreted by ETEC and further disrupt the electrolyte balance in the intestines, thus leading to diarrhea [2,3]. Inhibition of the proliferation of bacteria and the production of enterotoxins are effective strategies to cope with the ETEC challenge. Antibiotics are the most popular treatment in response to ETEC infection; however, in livestock production, the abuse of antibiotics and antimicrobial agents aggravates the occurrence of antibiotic-resistant bacteria, which further transfers them to human pathogens [4]. Therefore, it is urgent to develop green feed additives to protect humans’ and farm animals’ intestinal health.

Medium-chain fatty acids (MCFAs) contain 6–12 carbon atoms and have received extensive attention for their numerous biological functions, such as energy substrates and antibacterial and immune-regulation activity [5]. MCFAs inhibit the pathogenicity of pathogens by promoting MCFAs’ transportation into the cytoplasm and, thus, the leakage of intracellular substances [6]. Lauric acid, the most active saturated fatty acid in coconut oil, is considered a safe and effective antibiotic substitute because of its strong antibacterial, antiviral, and anti-inflammatory properties [7,8,9]. Lauric acid has shown antibacterial activity against Gram-positive and Gram-negative pathogens, such as *Staphylococcus aureus*, *Escherichia coli*, *Helicobacter pylori*, and *Streptococcus mutans* [10]. However, MCFAs are not easily accepted by livestock and poultry because of their unpleasant smell; thus, they generally exist as medium-chain triglycerides in foods and animal feeds [11]. Lauric acid also exhibits a strong chelating capacity, for example, Ca^2+^, which can promote intestinal absorption [12]. Zinc also plays a vital role in the growth, immunity, and mucous membranes through participation in the synthesis of nucleic acids and proteins, energy metabolism, and synthesizing a variety of important metabolic enzymes [13]. Our previous study demonstrated that zinc alleviated diarrhea induced by ETEC in a virulence-factors-dependent manner [14]. However, the low availability of zinc in an inorganic form limits its use in practice; this prompted us to develop organic forms of zinc with a higher bacterial inhibitory capacity. Therefore, chelating zinc with lauric acid is a promising strategy to enhance its anti-diarrhea function in response to ETEC infection.

ZnLa mainly exists in the form of two molecules of lauric acid chelating with one molecule of zinc ion. Our previous study demonstrated that ZnLa has potent antibacterial activity against ETEC [15]. However, limited information regarding the regulatory role of ZnLa in vivo was acquired. In addition, high levels of zinc in the diet could be toxic to animals, as high levels of zinc result in oxidative stress, reduced appetite, and even death [16]. Therefore, the present study aimed to investigate the effect of ZnLa on the intestinal barrier and inflammation with the goal of avoiding the potential toxic effect of zinc in a mouse model of diarrhea induced by ETEC infection and marginal zinc deficiency (dZn). In the present study, we established the mice diarrhea model and the dZn model via oral gavage ETEC and a feeding dZn diet, respectively, and further explored the effect of ZnLa on host intestinal function and its mechanisms.

## 2. Materials and Methods

### 2.1. Statement of Ethics

All procedures on mice were conducted in accordance with the “Guiding Principles in the Care and Use of Animals” (China) and were approved by the Laboratory Animal Ethics Committee of Biological Engineering, Henan University of Technology (ethical approval code: HUT202005-1).

### 2.2. Experimental Design

In the present study, female mice were used to establish the ETEC-infected diarrhea model according to previous studies [17,18,19]. Female ICR mice, specific-pathogen-free (SPF) and 3 weeks old, were provided by Sibeifu Biotechnology company (Beijing, China). All mice adapted to the new laboratory environment for seven days before the start of the animal experiment. The mice were raised at the same temperature (22 ± 2 °C) and humidity (50 ± 5%) in a control room and a 12-h light/dark cycle under specific-pathogen-free conditions and adapted to the new laboratory environment for seven days before the experiment. Food and water were ad libitum during the experiment unless specifically stated. ZnLa was purchased from Macklin (Shanghai Macklin Co., Shanghai, China) with a purity above 98%.

The first experiment aimed to investigate the protective effect of ZnLa on the intestinal health of mice with ETEC infection; a total of 45 ICR female mice were randomly divided into a dZn group (dZn); dZn and ETEC challenge group (dZn+ETEC); and ETEC infection together with low (200 mg/kg ZnLa), middle (400 mg/kg ZnLa) or high (600 mg/kg ZnLa) dosage of ZnLa groups (ZnLa LOW+ETEC, ZnLa MID+ETEC, ZnLa HIGH+ETEC). Mice in both dZn and dZn + ETEC groups were fed a dZn diet, and mice in ZnLa LOW+ETEC, ZnLa MID+ETEC, and ZnLa HIGH+ETEC groups were fed a dZn diet with 200 mg/kg, 400 mg/kg, and 600 mg/kg ZnLa, respectively, and zinc levels were 35.4 mg/kg, 64.8 mg/kg, and 95.2 mg/kg, respectively. AIN-93G dZn feed (S10033G) was purchased from Jiangsu Xietong Pharmaceutical Bio-engineering Company (Nanjing, China), which contained 7.0 mg/kg zinc. The composition and nutritional levels of the dZn diet are shown in Appendix A.

The second experiment was designed to compare the effects of ZnLa and ZnSO_4_ with equal levels of zinc on the growth curves of ETEC in vitro. According to previous research [20], the zinc levels in the culture fluid were 30 and 15 mg/L, respectively; the details of the method are shown in Section 2.10.

To further compare the effect of ZnLa and ZnSO_4_ with equal amounts of zinc on ETEC inhibition in vivo, we analyzed the ETEC shedding and virulence factors of mice in the third experiment. A total of 36 mice were randomly divided into a ZnLa group (ZnLa), a ZnLa and ETEC infection group (ZnLa+ETEC), a ZnSO_4_ group (ZnSO_4_), and a ZnSO_4_ and ETEC infection group (ZnSO_4_+ETEC). The diet zinc content of the ZnLa group and the ZnSO_4_ group were 30 mg/kg, adopted from the mice treated with ZnLa LOW in the first experiment.

The ETEC infection was performed in accordance with our previous study [14]. Briefly, mice were treated with streptomycin (5 g/L) and fructose (67 g/L) in drinking water 36 h before the ETEC infection. All mice fasted for 12 h and were given intraperitoneal cimetidine (50 mg/L, American Sigma) 1–2 h before the ETEC infection. On the 16th day of the experiment, all mice in the ETEC-challenged group were orally gavaged with ETEC 1 × 10^9^ CFU, and the mice in the non-infected group were fed an equal dose of culture medium. The ETEC serotype was O149:K91:K88ac. The fresh feces and jejunum tissues of mice were collected 48 h after the ETEC infection.

### 2.3. Samples Collection

The mice were weighed every 12 h. The consistency of the stool was recorded at 24 h and 48 h after the attack. Stools with a normal shape, soft without shape, loose, or liquid (translucent), or blood were recorded as scores of 1, 2, 3, and 4, respectively.

All mice were anesthetized and sampled after 48 h of ETEC inoculation. Blood samples of mice were collected from the eyes, and the serum was stored in a −20 °C refrigerator for further analysis. The liver, kidney, spleen, and small intestine were quickly removed, washed with a cooled phosphate-buffered solution, and weighed. The jejunum was separated from the small intestine and stored in a 4% paraformaldehyde tissue fixation solution (Seville Biotechnology Co., Ltd, Wuhan, China.) or liquid nitrogen. Samples of jejunal tissue and cecal chyme were collected, stored in liquid nitrogen for 2 h, and further stored in a −80 °C refrigerator for follow-up experiments. The additional jejunal tissue sample was collected and washed with ice-cold normal saline to remove the intestinal contents.

### 2.4. Determination of Zinc Content

Measurement of zinc content in diets using flame atomic absorption spectrometry (ContrAA, Analytik Jena, Jena, Germany).

### 2.5. Organ Index

The liver, kidney, and spleen of each mouse were weighed. The relative weight of each organ was calculated according to the following formula:Organ weight index = organ weight/body weight × 100

### 2.6. Intestinal Morphology

Jejunal tissues were fixed in 4% paraformaldehyde buffer (Solarbio, Beijing, China) for more than 48 h, dehydrated, and placed in transparent treated xylene. After paraffin soaking and embedding, the transparent jejunum was cut into 5 μm tissue specimens on a paraffin slicer and stained with hematoxylin and eosin. The villi were observed under a microscope, and the microstructure of the jejunum was analyzed.

### 2.7. Inflammatory Cytokines 

According to the standard procedure described by the manufacturer (Nanjing Jian Cheng Co., Ltd., Nanjing, China), the levels of serum interleukin-1β (IL-1β), interleukin-6 (IL-6) and tumor necrosis factor-α (TNF-α) were measured by the mouse ELISA kit. The standard curve was drawn by ELISACalc regression fitting calculation software V0.1.

### 2.8. Intestinal Permeability

Intestinal permeability was evaluated by measuring the concentrations of serum diamine oxidase (DAO), D-lactic acid (DLA), and endotoxin. The specific determination methods of serum DAO and DLA indexes were determined based on the instructions of the ELISA kit (Nanjing Jian Cheng Co., Ltd., Nanjing, China). The Limulus endotoxin test kit (Xia Men Bioendo Technology Co., Ltd., Xiamen, China) was used to determine the content of serum endotoxin. The concentrations of serum DAO, DLA, and endotoxin were calculated according to the standard curve.

### 2.9. Bacterial Load

The bacterial load was measured as previously described [17]. About 100 mg of fresh feces and jejunal tissues from mice were weighed in a sterile tube. After homogenization using a homogenizer (Iika, Germany), they were evenly plated on MacConkey agar medium (OXOID, Hampshire, UK) with gradient dilution. After incubation at 37 °C for 24 h, the cell-colony-forming units (CFU) were counted. The results were expressed as log10 CFU/g feces or jejunum tissue.

### 2.10. Effect of Different Source of Zinc on the Growth Curve of ETEC In Vitro

ETEC was used with ZnSO_4_ or ZnLa to create the final zinc concentrations of 30 and 15 mg/L, respectively. The DMEM cultural medium was used as the blank control group. The ETEC was cultured at 37 °C with 180 r/min to maintain the rapid growth of ETEC and sampled at 0, 2, 4, 6, 8, 10, 12, 14, and 16 h, respectively. The growth curve was drawn by the incubation time and OD600 nm value.

### 2.11. Gene Expression

All primers used in the experiment were designed by the Primer BLAST procedure of NCBI, and its detailed sequences are shown in Appendix A. Total RNAs were extracted from jejunum tissue with Trizol reagent (Vazyme, Nanjing, China). NanoDrop 2000 (Thermo Fisher Scientific, Waltham, MA, USA) was used to evaluate the purity and concentration of RNA. The purity of RNA was verified by the absorbance ratio at 260 nm/280 nm. Complementary DNA (cDNA) was synthesized according to the manufacturer’s instructions with a reverse transcription kit (Vazyme, Nanjing, China). Next, a real-time quantitative polymerase chain reaction (RT-qPCR) was performed by quantitative fluorescence PCR (Analytik Jena, Jena, Germany) and quantified by SYBR Green qPCR Master Mix (Vazyme, Nanjing, China). A 20 μL reaction system, including a 2 μL cDNA template, 0.4 μL reverse primer (10 μM), 0.4 μL forward primer (10 μM), 7.2 μL dd H_2_O, and 10 μL ChamQ Universal SYBR qPCR Master (2×), was constructed. The amplification condition was the following: 95 °C for 30 s, 40 cycles of 95 °C for 10 s, and 60 °C for 30 s, and a melt curve analysis with 95 °C for 15 s, 60 °C for 60 s, and 95 °C for 15 s. In this study, GAPDH was used as the housekeeping gene. The relative expression of target genes was determined by the 2^−ΔΔCT^ method.

### 2.12. Virulence Factors Assay

All RT-PCR gene-specific primers for virulence factors are listed in the attached Appendix A. The mRNA expression of genes related to virulence factors in cecum contents was measured the same as that in the jejunum mRNA. In Escherichia coli, GAPDH is produced by GAPA. The relative expression level of cecal contents was calculated by the 2^−ΔΔCT^ method, with GAPA as the internal control [21].

### 2.13. Statistical Analysis

All data in the experiment were analyzed using a two-way ANOVA of GraphPad Prism (version 8.0.2), and a Newman-Keuls test was used to compare the means between different treatment groups. All data are expressed as means ± sem. *p* < 0.05 indicates that they are statistically significant.

## 3. Results

### 3.1. Effects of ZnLa on the Clinical Symptoms of Mice with ETEC Challenge

ETEC infection aggravated body weight losses, organ index, and diarrhea scores in mice compared with their non-infected counterparts, which indicates that the mice diarrhea model was successfully established. However, the body weight losses at 36 h and 48 h post-infection were lower (*p* < 0.05) in ZnLa LOW+ETEC, ZnLa MID+ETEC, and ZnLa HIGH+ETEC groups than the dZn+ETEC group (Figure 1a), though the average daily zinc intake was not different (*p* > 0.05) in the dZn, dZn+ETEC, and ZnLa Low+ETEC groups (Appendix A). In addition, the liver, kidney, and spleen indexes were lower (*p* > 0.05) in the Znla LOW+ETEC, Znla MID+ETEC, and Znla HIGH+ETEC groups than the dZn+ETEC group (Figure 1d–f). The diarrhea scores were also lower (*p* < 0.05) at 24 and 48 h post-infection in mice with ETEC infection after feeding with the ZnLa diet than the dZn+ETEC group (Figure 1b,c). 

### 3.2. Effect of ZnLa on the Intestinal Morphology of Mice with ETEC Infection

The jejunum is the main organ accounting for nutrient absorption. The morphological and histopathological changes of jejunal villi were determined by hematoxylin and eosin staining. Histopathology damage in mice with ETEC challenge was prevented when pretreated with ZnLa, as manifested by the higher (*p* < 0.05) villus height and villus height/crypt depth in the ZnLa LOW+ETEC, ZnLa MID+ETEC, and ZnLa HIGH+ETEC group than the dZn+ETEC group (Figure 2a–d).

### 3.3. Effect of ZnLa on Intestinal Barrier Damage Caused by ETEC Infection

DAO and DLA in serum are commonly used to judge intestinal barrier function. The serum levels of DAO, DLA, and endotoxin in the dZn+ETEC group were significantly higher (*p* < 0.05) than those in the dZn group (Figure 3a–c). However, the serum levels of DAO, DLA and endotoxin were lower (*p* < 0.05) in the ZnLa LOW+ETEC, ZnLa MID+ETEC, and ZnLa HIGH+ETEC groups than the dZn+ETEC group.

To further explore the effect of ZnLa on the intestinal barrier function of mice with ETEC infection, we measured the relative expression of genes involved in the intestinal tight junction. Oral administration of ETEC remarkably decreased (*p* < 0.05) the mRNAs expression of *occludin*, *Muc-2*, and *ZO-1* in the jejunum (Figure 3d,f,g), whereas ZnLa pretreatment prevented (*p* < 0.05) the decrease in the mRNA expression of *occludin* with ETEC challenge (Figure 3d). A decrease (*p* < 0.05) in mRNA expression of *ZO-1* in the ZnLa MID+ETEC and ZnLa HIGH+ETEC groups was also prevented, compared with the dZn+ETEC group (Figure 3g). 

### 3.4. Effect of ZnLa on Intestinal Inflammatory Reaction in ETEC-Challenged Mice

In addition to permeability and action as an intestinal barrier, ZnLa pretreatment also protected against intestinal inflammation in ETEC-challenged mice. The protein and relative mRNAs levels of *IL-1β*, *IL-6* and *TNF-α* were higher (*p* < 0.05) in the dZn+ETEC group than in the dZn group (Figure 4a–f). However, pretreatment with ZnLa protected mice against the intestinal inflammatory responses induced by ETEC infection, as manifested by the lower (*p* < 0.05) serum levels of IL-1β and TNF-α and mRNA expression of jejunal *IL-6* and *TNF-α* in the ZnLa LOW+ETEC, ZnLa MID+ETEC and ZnLa HIGH+ETEC groups than the dZn+ETEC group (Figure 4a,c,e,f). Simultaneously, the increase (*p* < 0.05) in serum IL-6 levels in the ZnLa MID+ETEC group and the mRNA expression of jejunal *IL-1β* in the ZnLa MID+ETEC and ZnLa HIGH+ETEC groups were prohibited, compared with the dZn+ETEC group (Figure 4b,d).

### 3.5. Effect of ZnLa on the Toll-like Receptor 4 (TLR4)/Myeloiddifferentiation Factor 88 (MYD88)/Nuclear Factor κB (NF-κB) Signaling Pathway in Mice with ETEC Challenge

The relative expression of *GRP39*, a zinc receptor, in the jejunum was decreased (*p* < 0.05) in the dZn+ETEC group compared to the dZn group, whereas the mRNA expression of *GPR39* was higher (*p* < 0.05) in the ZnLa MID+ETEC and ZnLa HIGH+ETEC groups than the dZn+ETEC group (Figure 5a).

To clarify whether ZnLa prevented intestinal inflammation through the *TLR4/MYD88/NF-κB* signaling pathway, the related mRNA expressions in the jejunum were measured. The result showed that the ETEC infection significantly increased (*p* < 0.05) the relative expression of *TLR4*, *MYD88*, and *NF-κB* in the jejunum; however, the increase in mRNA expression of *TLR4, MYD88,* and *NF-κB* in the jejunum was prevented (*p* < 0.05) by pretreatment with ZnLa before ETEC challenge (Figure 5b–d).

### 3.6. Effect of ZnLa on Anion Transporters in Mice Challenged with ETEC

ETEC infection likely decreased the ability of Na^+^ absorption and chloride ion transport in the intestine. The relative expression level of Na^+^/H^+^ exchanger 3 *(NHE3)* in the jejunum was decreased (*p* < 0.05), whereas the relative expression level of the cystic fibrosis transmembrane transport regulator *(CFTR)* was increased (*p* < 0.05) in the dZn+ETEC group compared with the dZn group. However, the decrease in mRNA expression of *NHE3* and the increase in mRNA expression of *CFTR* were prevented (*p* < 0.05) by pretreatment with ZnLa before the ETEC challenge (Figure 6a,b).

### 3.7. Effect of ZnLa on ETEC Shedding in the Stool and Intestine

ETEC colonization plays a vital role in the pathogenesis of ETEC infection. The concentration of ETEC in the feces and jejunum was significantly higher (*p* < 0.0001) in the dZn+ETEC group than in the dZn group. The concentration of ETEC in the feces of the ZnLa MID+ETEC group and the concentration of ETEC in the jejunum of the ZnLa LOW+ETEC, ZnLa MID+ETEC, and ZnLa HIGH+ETEC groups were lower (*p* < 0.05) than the dZn+ETEC group (Figure 7a,b).

### 3.8. Effect of ZnLa on the Virulence Factors of Intestinal Content in ETEC-Infected Mice

To further investigate the underlying mechanism of intestinal damage and ETEC shedding, the relative expression of genes involved in ETEC virulence factors in cecum content was measured. Compared to the dZn group, the mRNA expression levels of mRNAs involved in LT (eltA and eltB), ST (estB), biofilm formation (bssS), motility (Mota), and cellular adhesion (FaeG) in cecum content were significantly increased (*p* < 0.05) in the dZn+ETEC group. The increase in mRNA expression of *eltA, estB, bssS,* and *faeG* was dramatically hindered (*p* < 0.05) by pretreating with ZnLa before ETEC challenge, though no significant effect on the relative expression of mRNAs was involved in quorum sensing (LuxS) and biofilm formation (tnaA). It is worth noting that the relative expression of *eltB* was also lower (*p* < 0.05) in the ZnLa LOW+ETEC and ZnLa MID+ETEC groups than in the dZn+ETEC group (Figure 8a–h).

### 3.9. Effects of Different Zinc Sources on ETEC Inhibition In Vitro and Vivo

To investigate the antibacterial capacity of ZnLa on ETEC, we further compared the inhibitory roles of ZnLa and inorganic zinc-ZnSO_4_ in vivo and in vitro. ZnLa and ZnSO_4_ both inhibited the growth of ETEC with increasing concentrations in vitro. However, the growth of ETEC was completely inhibited by ZnLa at a zinc concentration of 30 mg/L but not ZnSO_4_ (Figure 9a)_._ These observations indicated that ZnLa exhibited a stronger inhibitory effect on ETEC than ZnSO_4_. Consistent with results in vitro, the concentrations of ETEC in mouse feces and jejunum showed a similar change trend as in vitro, and the average daily zinc intakes were not different (*p* > 0.05) among the ZnSO_4_, ZnLa, ZnSO_4_+ETEC, and ZnLa+ETEC groups (Appendix A). The concentrations of ETEC in feces and jejunum were lower (*p* < 0.05) in the ZnLa+ETEC group than the ZnSO_4_+ETEC group, though there was no significant difference (*p* > 0.05) between the ZnSO_4_ and ZnLa groups (Figure 9b,c). 

### 3.10. Effects of Different Zinc Sources on the Relative Expression of Virulence Genes

Consistent with the above observations, there was no significant difference (*p* > 0.05) in the relative expression of virulence genes between the ZnSO_4_ and ZnLa groups in mice without ETEC infection. However, the relative expressions of *eltA, eltB, estB, bssS,* and *motA* were lower (*p* < 0.05) in the ZnLa+ETEC group than in the ZnSO_4_+ETEC group (Figure 10a–h).

## 4. Discussion

MCFA has attracted widespread attention, from human foods to animal feeds, for its role in inhibiting pathogenic bacteria and regulating blood glucose, lipid metabolism, and gut health [22,23]. Piglets pretreated with formic acid (600 mg/kg) and monolaurin (200 mg/kg) exhibited lower fecal score and rectal temperature after the ETEC challenge. [24]. Lauric acid at a dosage of 1000 mg/kg alleviated body weight loss and intestinal mucosal damage caused by LPS challenge in broilers [25]. However, limited information was acquired regarding the effect of MCFA chelated with zinc on ETEC infection. As expected, it was found that ETEC infection reduced the body weight of mice and that 200 mg/kg of ZnLa could effectively protect against body weight losses caused by ETEC infection in the present study. In addition, the increased diarrhea scores induced by ETEC infection were effectively prevented in mice pretreated with ZnLa. NHE3 and CFTR can mediate intestinal Na^+^ absorption and chloride ion transport into the intestinal tract, respectively [26,27]. Enterotoxin attaches to the surface of intestinal epithelial cells, activates adenylyl cyclase, and stimulates the secretion of water and electrolytes in the intestinal lumen through NHE3 and CFTR, thereby triggering diarrhea [28,29,30]. We also found that mRNA expression of *CFTR* was higher and *NHE3* was lower in mice with ETEC infection; however, ZnLa pretreatment prevented this trend and reduced diarrhea in mice. This study provides a new insight into how ZnLa regulates ETEC infection.

MCFAs can not only provide energy to intestinal cells but also participate in regulating gut health. Dietary supplementation with MCFAs improved the intestinal morphology, villus height, and crypt depth of weaned piglets [31,32,33]. Consistent with the above results, ZnLa prevented intestinal injury in this study by improving the morphology of the jejunum and maintaining the villus height and villus height/crypt depth of the jejunum. However, in a study on the metal chelate of lauric acid, it was found that adding calcium laurate to the diet of weaned piglets did not significantly improve the jejunum villus height [34]. This difference between ZnLa and calcium laurate might be related to the higher absorption capacity of ZnLa in the proximal small intestine and thereafter the protective function of zinc.

The intestinal barrier is an essential functional barrier against pathogens and toxins, and increased intestinal permeability is often associated with intestinal damage [35]. Zinc is also involved in the repair of intestinal physical barrier damage by improving the relative expression of *occludin*, *ZO-2,* and *ZO-3* after ETEC infection [36]. It has been proven that MCFAs protect intestinal barrier function by regulating the expression of the tight junction protein [37,38]. In this study, the plasma concentrations of DAO, DLA, and endotoxin were higher in the dZn+ETEC group, whereas these increments were prevented by ZnLa pretreatment. In addition, intercellular tight junction proteins play an essential role in maintaining intestinal barrier integrity and intestinal permeability [39,40]. For example, ZO-1 is beneficial to the repair of intestinal mucosal injury [41], while occludin is considered to be a regulator of tight junction assembly and function [42]. It has been found that oral administration of lauric acid protected the intestinal barrier damage induced by deoxynivalenol by upregulating the expression of *Occludin*, *Claudin-1,* and *ZO-1* in the jejunum of mice [43]. In the present study, ZnLa improved the host’s defense against intestinal barrier damage, which was associated with the preventative role of ZnLa in the decreased expression of *Occludin* and *ZO-1* in the jejunum induced by ETEC infection. GPR39, as a zinc-sensing receptor that is ubiquitously expressed throughout the gastrointestinal tract, has dual roles in promoting intestinal epithelial cell proliferation and tight junction protein expression [44]. We also found that the relative expression of *GPR39* in mice with ETEC infection increased with increasing zinc intake. These observations suggested that the protective role of ZnLa on the intestinal barrier may be related to both Lauric acid and zinc.

Endotoxins, produced by ETEC, can cause intestinal inflammation through the TLR4/MyD88/NF-κB signaling pathway [45]. The anti-inflammatory effects of medium-chain triglycerides and lauric acid have been confirmed in previous studies [7,46,47,48]. It has been proven that formic acid and monolaurin pretreatment protected against the increase of plasma TNF-α, IL-6, and IL-1β and the mRNA expression of TNF-α, IL-1β, and IL-6 in the piglets after the ETEC challenge [24]. Consistent with previous studies, the relative expression of *TLR4*, *MyD88*, and *NF-κB* in the jejunum, and the resultant expression of proinflammatory cytokines IL-1β, IL-6, and TNF-α were higher in ETEC-challenged mice. Notably, we noticed that ZnLa pretreated impeded the increased in *TLR4*, *MyD88*, *NF-κB*, and thereafter IL-1β, IL-6 and TNF-α in mice with ETEC infection. These results implicated the protective role of ZnLa in intestinal inflammation.

ETEC can further increase intestinal permeability and cause inflammation by colonizing intestinal epithelial cells and secreting enterotoxins, which destroy the intestinal barrier function [49]. MCFAs (caprylic acid, decanoic acid, and lauric acid) combined with edible plant essential oils could accelerate the transport of antibacterial compounds into the cytoplasm and the uptake of hydrogen ions from organic acids in *Escherichia coli*, thus leading to bacterial death [50]. Similarly, our previous experiments in vitro proved that ZnLa inhibited the proliferation of ETEC by changing the permeability of cell membranes and increasing the leakage of nucleic acid proteins in cells. To further verify the inhibitory effect of ZnLa on *ETEC* in vivo, we analyzed the density of *ETEC* in feces and jejunum of mice and found that ZnLa improved gut health of the host by hindering the increased concentration of ETEC colonization in feces and intestine, and the inhibitory effect was better than that of ZnSO_4_ in the same concentration of zinc.

The pathogenicity of ETEC depends on host-specific fimbrial adhesins and enterotoxin secretion. The main component of fimbriae is FaeG, which can help F4 fimbrial adhesin combine with the intestinal cell target and activate its pathogenicity [51,52]. At the same time, the pathogenicity of ETEC is based on the secretion of LT and ST in the intestinal epithelium, which are the main causes of diarrhea symptoms [3]. Previous studies have demonstrated that dZn can obviously upregulate the mRNA expression of virulence genes in the cecum contents of ETEC infection mice [14]. In the present study, the increased expression of genes involved in cellular adhesion (FaeG), LT (eltA and eltB), ST (estB), and biofilm formation (bssS) in the cecum of mice induced by ETEC infection was prevented by pretreatment with ZnLa. The flagella movement is also crucial for the colonization of ETEC in the hindgut. In this study, only a high dose of ZnLa prevented the upregulation of the level of motility (motA) gene expression during ETEC infection. These observations might in part account for the protective role of ZnLa in ETEC colonization, the intestinal barrier, and the inflammation response in the intestine.

Remarkably, the protective function of ZnLa in response to ETEC in the present study was established in the ETEC infection and marginal zinc deficiency model, which was primarily aimed to avoid the potential toxicity of zinc. However, zinc deficiency has been associated with neurodegenerative diseases and increased intestinal permeability [53], which may limit its practicality in animals fed normal zinc levels. Further research is required to explore the role of ZnLa under different physiological and inflammatory conditions and zinc levels in animal models.

## 5. Conclusions

In conclusion, ZnLa protected against the intestinal barrier injury and inflammation caused by ETEC infection in mice under the condition of dZn. The protective function of ZnLa on ETEC-infected mice was highly correlated with its regulatory role on virulence factors and zinc receptor-*GPR39*. This study showed that ZnLa was a more promising strategy for protecting against ETEC infection compared to inorganic zinc.

## Figures and Tables

**Figure 1 nutrients-15-00054-f001:**
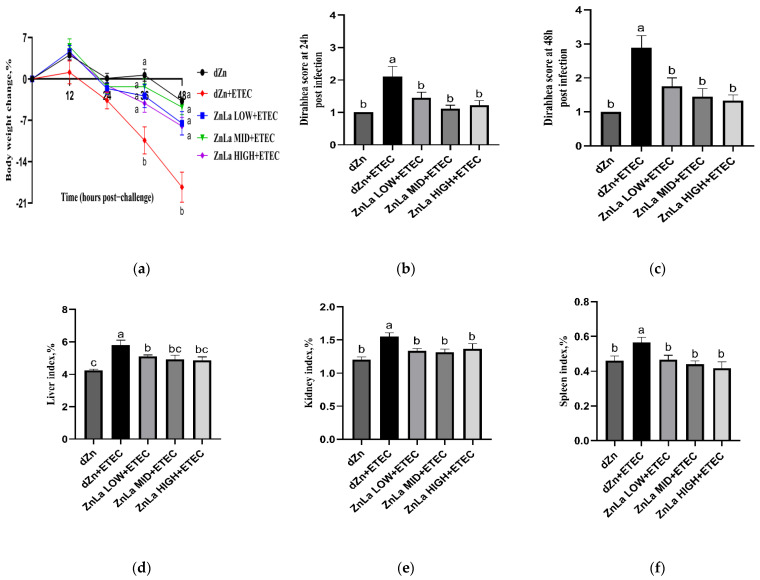
Effects of ZnLa on the clinical symptoms in mice with ETEC challenge. (**a**) Body weight change from 0 to 48 h. (**b**,**c**) The diarrhea score at 24 and 48 h. (**d**–**f**) The liver index, kidney index, and spleen index. The data are expressed as mean ± s.e.m (n = 9), and different lowercase letters in each group show significant differences (*p* < 0.05).

**Figure 2 nutrients-15-00054-f002:**
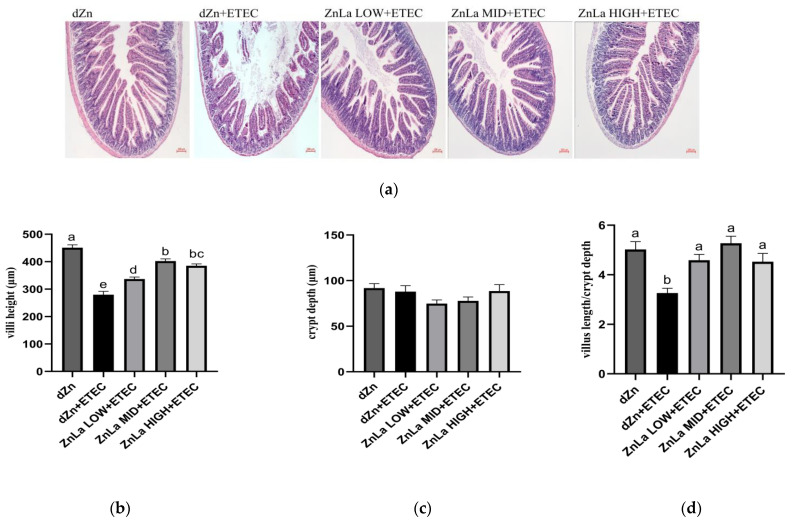
Effects of ZnLa on the jejunum morphology of mice with ETEC challenge. (**a**) Jejunum structure (×100). (**b**–**d**) The (**b**) villus height, (**c**) crypt depth, and (**d**) villus height/crypt depth in the jejunum. The data are expressed as mean ± s.e.m (n = 9), and different lowercase letters in each group show significant differences (*p* < 0.05).

**Figure 3 nutrients-15-00054-f003:**
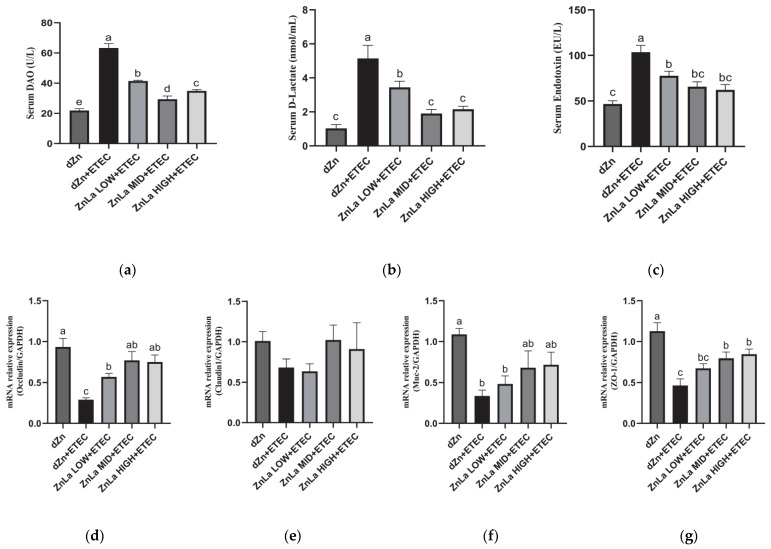
Effect of ZnLa on the intestinal barrier function of mice with ETEC challenge. Serum levels of (**a**) DAO, (**b**) DLA, and (**c**) Endotoxin. Relative mRNA expression of (**d**) *Occludin*, (**e**) *Claudin-1*, (**f**) *MUC-2*, and (**g**) *ZO-1* in the jejunum. The data are expressed as mean ± s.e.m (n = 9), and different lowercase letters in each group show significant differences (*p* < 0.05).

**Figure 4 nutrients-15-00054-f004:**
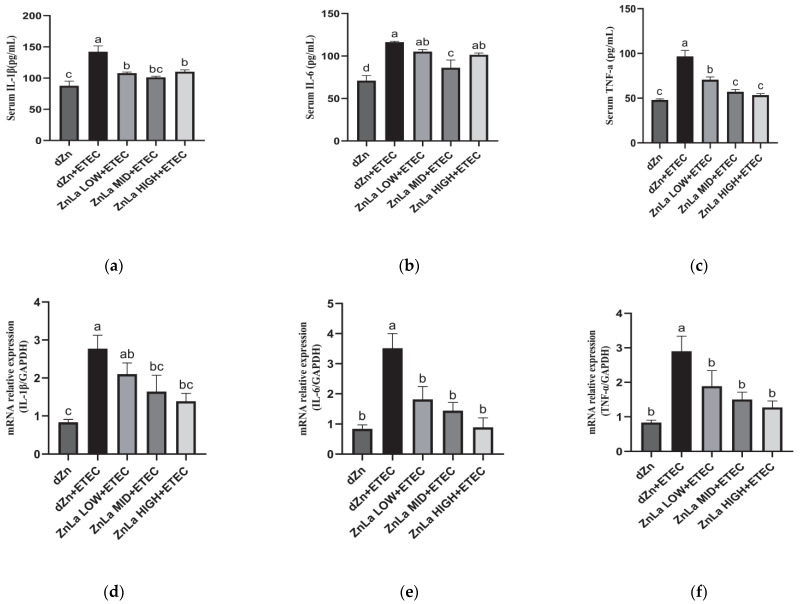
Effects of ZnLa on intestinal inflammation of mice with ETEC challenge. Serum levels of (**a**) IL-1β, (**b**) IL-6, and (**c**) TNF-α. Relative mRNA expression of (**d**) *IL-1β*, (**e**) *IL-6*, and (**f**) *TNF-α* in the jejunum. The data are expressed as mean ± s.e.m (n = 9), and different lowercase letters in each group show significant differences (*p* < 0.05).

**Figure 5 nutrients-15-00054-f005:**
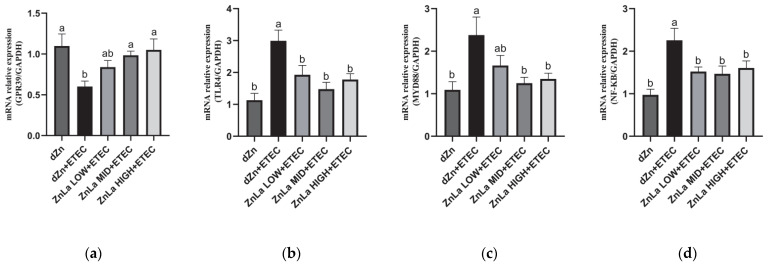
Effect of ZnLa on TLR4/MYD88/NF-κB signaling pathway in mice with ETEC challenge. Relative mRNA expression of (**a**) *GPR39*, (**b**) *TLR4*, (**c**) *MYD88*, and (**d**) *NF-κB* in the jejunum. The data are expressed as mean ± s.e.m (n = 9), and different lowercase letters in each group show significant differences (*p* < 0.05).

**Figure 6 nutrients-15-00054-f006:**
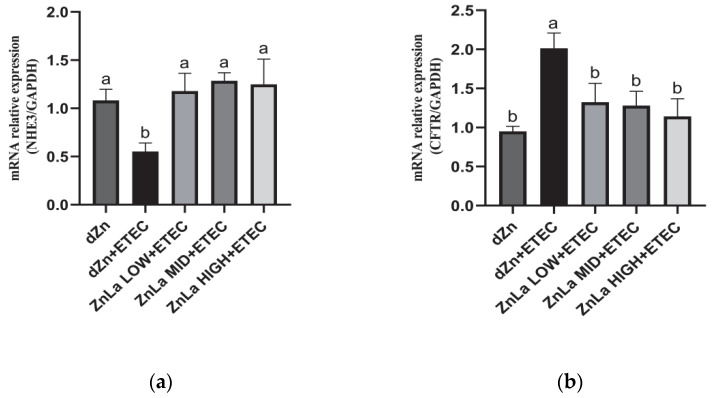
Effect of ZnLa on ion absorption and secretion in mice challenged with ETEC. Relative mRNA expression of (**a**) *NHE3* and (**b**) *CFTR* in jejunum. The data are expressed as mean ± s.e.m (n = 9), and different lowercase letters in each group show significant differences (*p* < 0.05).

**Figure 7 nutrients-15-00054-f007:**
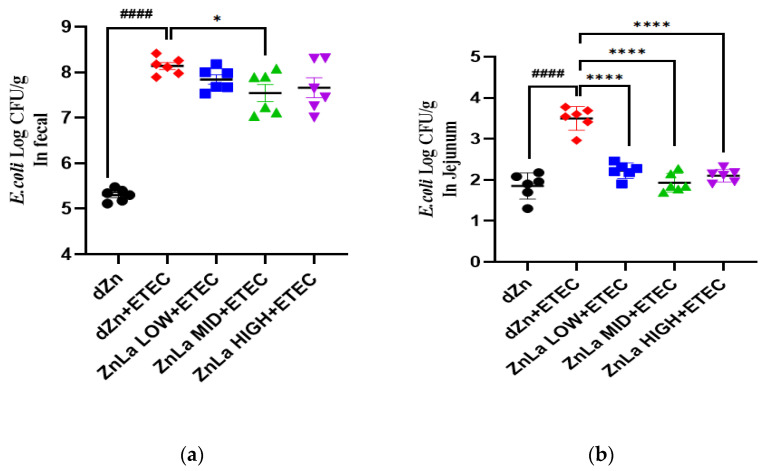
Effect of ZnLa on the content of ETEC in the feces and jejunum with ETEC infection mice. (**a**) ETEC content in feces; (**b**) ETEC content in the jejunum. The data are expressed as mean ± s.e.m (n = 6). (#### represent *p* < 0.0001 in dZn compared with dZn+ETEC. *, **** represent *p* < 0.05, *p* < 0.0001 in dZn+ETEC compared with ZnLa+ETEC).

**Figure 8 nutrients-15-00054-f008:**
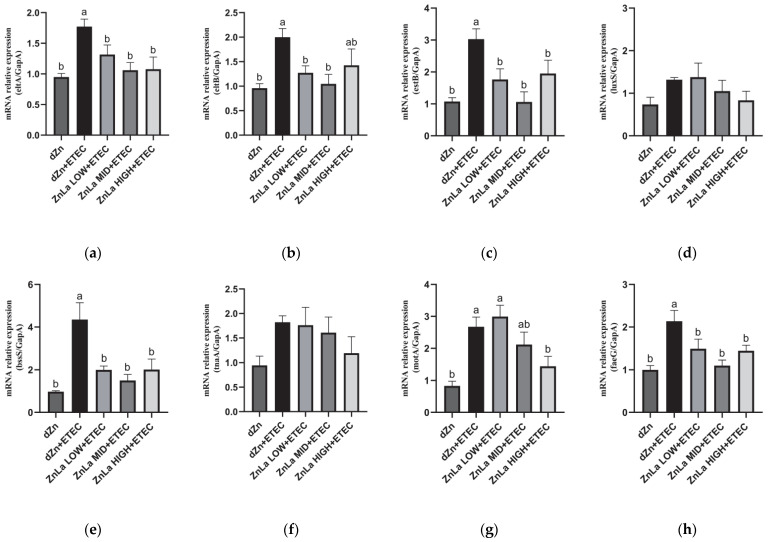
Effects of ZnLa on the relative expression of virulence factors in the cecal content of mice with ETEC infection. (**a**,**b**) LT (eltA and eltB), (**c**) ST (estB), (**d**) quorum sensing (luxS), (**e**,**f**) biofilm formation (bssS and tnaA), (**g**) motility (motA), and (**h**) cellular adhesion (faeG). The data are expressed as mean ± s.e.m (n = 9), and different lowercase letters in each group show significant differences (*p* < 0.05).

**Figure 9 nutrients-15-00054-f009:**
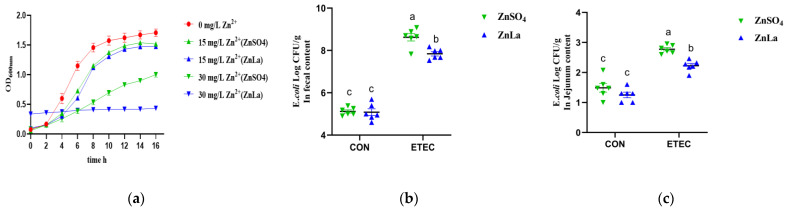
Effects of different zinc sources on the ETEC growth curve in vitro and ETEC shedding in mice. (**a**) ETEC growth curve and (**b**,**c**) content of ETEC in the feces and jejunum. The data are expressed as mean ± s.e.m (n = 6), and different lowercase letters in each group show significant differences (*p* < 0.05).

**Figure 10 nutrients-15-00054-f010:**
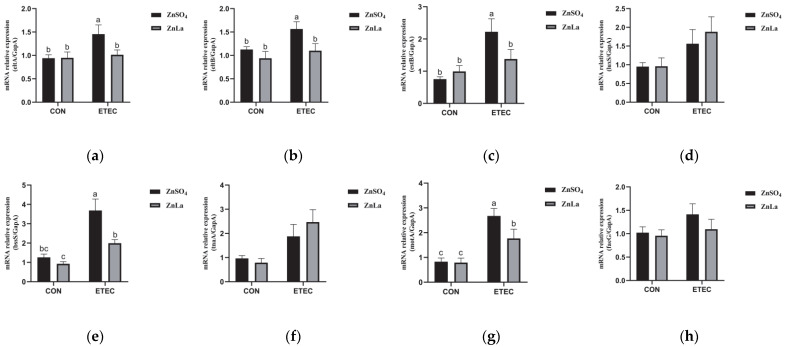
Effects of different zinc sources on virulence gene expression in the cecum content. (**a**,**b**) LT (eltA and eltB), (**c**) ST (estB), (**d**) quorum sensing (luxS), (**e**,**f**) biofilm formation (bssS and tnaA), (**g**) motility (motA), and (**h**) cellular adhesion (faeG). The data are expressed as mean ± s.e.m (n = 9), and different lowercase letters in each group show significant differences (*p* < 0.05).

## Data Availability

Not applicable.

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
