# Peer review of "Zinc Laurate Protects against Intestinal Barrier Dysfunction and Inflammation Induced by ETEC in a Mice Model"

_nutrients, 2022, doi:10.3390/nu15010054_

Round 1
Reviewer 1 Report
The work shows a promising effect of zinc laurate against intestinal membrane damage caused by ETEC inflammation model . The results are well- presented, the discussion is well-founded and the conclusion is in agreement. I wonder if it wouldn't be important to demonstrate the effect on a non-Zinc-deficient animal,the control group is a zinc deficient animal and I would like to see the effects on a non-deficient animal and be able to compare the effect of zinc laurate with data from a normal animal. Or would the proposal be a treatment only for zinc deficiency situations. If so, I suggest that an association between zinc deficiency and neurodegenerative diseases and the involvement of increased intestinal permeability should be included in the discussion. I suggest creating a graphical abstract for the work.in section 2.5 of the methods I suggest removing the word immune before organ in the index calculation formula
Author Response
Dear reviewer:
Please see the attachment.

Reviewer 2 Report
The objective of this investigation (lines 77-79) was to determine, in a mouse model, the effectiveness of dietary zinc laurate (ZnLa) in protecting against diarrhea and its associated intestinal pathologies induced by infection with an enterotoxin-producing strain of E. coli (ETEC). It appears that three separate experiments were performed, the first two in vivo and the third in vitro – but this is quite difficult to ascertain, in particular because of the confusing manner in which the Methods section is constructed (item 5 of Major Concerns, below). The studies conducted in vivo used recently-weaned (4 weeks old) female mice, ICR strain, fed test diets for 16 days and then orally challenged with ETEC for study 24 and 48 hours later. The first experiment was aimed at determining effects on body weight, severity of diarrhea and intestinal health indices, and included a comparison of three dietary levels of ZnLa. The second experiment was intended to compare the influence of dietary zinc (one dietary level, only) in laurate and sulphate forms on the expression of virulence factors of the chosen strain of ETEC. The third experiment (outlined in lines 169-173) was intended to compare the growth curves of ETEC when incubated in culture medium containing equal concentrations of zinc in either laurate or sulphate form. It is concluded that dietary ZnLa offered protection against ETEC-induced diarrheal disease, perhaps through inhibiting growth of the pathogen and inhibiting its expression of virulence factors (lines 437-441). Please accept the following comments as an attempt to improve the submission, in particular in terms of its clarity.
Major Concerns
1) It is unclear, within the manuscript, whether the findings pertaining to mechanisms underlying the health-related outcomes are being interpreted as demonstrating causation or are viewed as illustrations of association that may suggest causation but do not formally demonstrate it. Causation is stated explicitly in some locations (e.g., lines 395-396), whereas the more cautious assessment of association is stated equally explicitly in other locations (e.g., lines 29 and 438-440). Apart from this problem of inconsistency in the interpretation of the findings, it is necessary to recognize that the design of this work can reveal associations, but nothing more, between health-related outcomes and expression of cytokines and virulence factors. Causation can be inferred as a matter of speculation in the Discussion section (but nowhere else, and definitely not in the Abstract), but it must be absolutely clear that this is only speculation. Revision to achieve both clarity and rigour, therefore, is necessary here. Some suggestions as to a course of experimentation to establish causation also should be included if at all possible. [Further to this matter, please ensure that literature citations are, likewise, interpreted carefully with respect to the matter of causation, e.g., lines 385-387 and 395-397.]
2) A dose-dependent pattern of response to consumption of ZnLa is indicated several times (e.g., lines 24, 28, 208, 358 and possibly elsewhere) in relation to clinical signs of infectious pathology and associated cytokine expression. Figures 1 and 5, however, clearly show that the statistical analysis revealed no dose-dependent response over the range of ZnLa intakes tested because the three groups of mice receiving different dietary levels of ZnLa share a superscript letter. On what basis is a dose-dependent response perceived? Please note that visual inspection of the outcomes is an insufficient basis for disregarding the statistical analysis.
3) Lines 204-205: With apologies, this statement is misleading. The serum zinc concentration of the group consuming the lowest level of dietary ZnLa did not differ from concentrations reported for the two dZn groups (Figure S1).
4) Line 203: What, exactly, is meant by the statement, “dZn was also established”? Certainly there is no evidence of dietary zinc deficiency, even a marginal deficiency. The higher levels of serum zinc in the two groups consuming the higher levels of ZnLa presumably reflects a loss of ability to maintain a normal zinc concentration in the face of very high zinc intakes. This should be a point of discussion with respect to potential limitations to the practicality of this type of approach.
5) With apologies, the Methods section is so severely disorganized as to be obtuse. It is extremely difficult to determine exactly what was done in this piece of work. This is part of the reason for my rather long opening paragraph (above) summarizing my best guess as to the experiments conducted. Because of this problem, the Results section also emerges as a rather amorphous and confusing heap of findings a great many of which are difficult, at best, to connect with any particular experiment. Regardless of how interesting the findings may be, only the most determined reader is likely to devote time and energy attempting to untangle the presentation. This could be seen as a minor concern because it should be easy to address by simple editorial revision. I list it as a major problem because failure to address this matter is likely to neutralize the entire piece of work.
6) I note a very important matter of interpretation that is obscured because of incorrect and imprecise use of language. The design of this work can demonstrate a protective effect of ZnLa when this compound is consumed by animals that, subsequently, are subjected to ETEC challenge. This is entirely different from a therapeutic effect which is suggested when readers are told that ZnLa “attenuated” ETEC-induced injury (e.g., title), or “alleviated” an injury (e.g., lines 207, 277), or “reversed” an injury (line 240). Likewise, terms such as “increase”, “decrease” (e.g., subsections 3.3, 3.6, 3.7) and “up-regulated” (line 387) are misleading when, in fact, the data sets show that the selected indices have been “sustained”. Thus, “increases” and “decreases”, both, have been prevented (depending on the particular index of interest) by consumption of ZnLa before and during ETEC challenge. I have provided only a few examples, a small sample, of this problem for the sake of making the point. Suffice it to say that this is a very serious matter that must be addressed throughout the manuscript – beginning with its title.
7) Line 203-206 (and elsewhere): This statement could easily be understood (misunderstood??) to mean that a marginal zinc deficiency was achieved by feeding the dZn diet. However, the serum zinc concentrations shown in Figure S1 are high even for the dZN group. Serum zinc concentrations of laboratory mice normally lie within the range of 0.7 to as much as 1.2 mg/L. Certainly a level close to 2mg/L (dZn group, Figure S1) is far above the normal range – to the point of being a significant and worrisome anomaly. The same concern applies to the zinc concentrations shown in Figure S2, although there was no dZn group in the second experiment. A careful discussion of the high serum zinc concentrations shown in Figures S1 and S2 – but particularly on the part of the dZn group (Figure S1) – is needed within the text of the manuscript. [Please note item 1f of minor concerns, below. It may be necessary to delete the serum zinc measurements as methodologically flawed and, hence, misleading.]
8) Both main components of ZnLa could contribute to the apparently beneficial effects of the dietary chelate, and some allusion is made to this possibility within the manuscript. A focused and insightful assessment of independent and/or synergistic influences of zinc and lauric acid, to the extent that the design of this investigation permits, would add important substance to the manuscript.
Minor Concerns
1) In addition to item 5 of Major Concerns, above, some critical points of methodological detail are either not provided (but are needed within the text of the manuscript) or are provided in a vague, confusing way. Please note the following (not an exhaustive list of the problem):
a) Why was the animal model limited to females, only?
b) On what basis was a dietary zinc level of 30mg/kg chosen for the second experiment (lines 111-112)?
c) The dietary level of zinc in the dZn diet is not given in the text. More importantly, it is never made clear why there is concern that a zinc toxicity problem could arise (line 79), thereby necessitating use of a marginally zinc-deficient dietary base.
d) The chemical form of the micronutrients, both vitamins and minerals, must be specified in Table S1.
e) It is unclear, from the text, whether the groups given ZnLa in the first experiment were fed the dZn diet to which ZnLa had been added, or whether ZnLa was the only source of zinc in these diets.
f) What precautions were taken to minimize contamination of serum and diet samples used for assessment of zinc concentrations?
g) With regard to the assessment of jejunal bacterial load (Subsection 2.9), I presume that physiologically adherent bacteria, only, were of interest (but this is unclear). If so, how were the fragments of jejunal tissue prepared so as to eliminate non-adherent organisms?
h) Subsection 2.10: What was the basis for choosing zinc concentrations of 15mg/L and 30mg/L for the purpose of the experiment conducted in vitro? Also, what is “180r/min” and why was this methodologically important? A microbiologist will know, but many readers will only be left in confusion.
i) How many cycles were used in the PCR analyses?
j) line 193: What is GAPA? Does this refer to the E coli gapA gene that produces GAPDH? If so, then doesn’t this mean that the same housekeeping gene (except bacterial rather than mammalian) was used in the virulence factor assays as in the assays for other genes (lines 186-187)? Clarification is essential here.
2) With apologies and recognition of the extreme difficulties that can accompany presentation in a foreign language, it will be essential to revise under the guidance of somebody for whom English is a first language. Apart from countless examples of word choices and phraseology that render understanding difficult, there are numerous examples of statements that are important to an understanding of the manuscript, as a whole, but that defy unambiguous interpretation (e.g., lines 31, 173, 203-206, 221-224 and many more). [Items 5 and 6 in my list of major concerns highlight particularly important matters both of which probably arise, in significant measure, as a consequence of this problem of language inadequacy.]
3) Numerous literature citations are incomplete, rendering it difficult, at best, for readers to find them. I note citation numbers 2, 11, 14, 19, 30, 31, 35, 36, 38, 42, 44. Please check for other examples of this problem.
4) Please limit use of abbreviations to the minimum absolutely necessary. For example, neither ADFI (line 352) nor FCR (line 353) is needed, and I question whether VH (villus height) and CD (crypt depth) accomplish anything other than to force readers to hunt around within the text to remind themselves of the meaning. [Are HLT and HST (line 425) defined anywhere in the text?] A list of abbreviations is much needed in a prominent location, e.g. immediately preceding the Introduction section.
5) It would be helpful to eliminate irrelevant speculation from the Discussion section. As an example, I note lines 362-366.
Author Response

(The authors gave the same response as above.)

Round 2
Reviewer 2 Report
A significant effort has been made to revise the original submission and I take some pleasure in being able to say that, at least from my standpoint, the revised document is much improved over the original. Some points remain in need of attention and are itemized below.
1) Readers should be informed, within the text of the manuscript, the reason for restricting the work to female animals. The response to my original inquiry indicates that an established animal model was used in conducting this piece of work. That seems alright to me, at least at this stage, although eventually it will be necessary to expand the research to include males. The point here, however, is that readers need to be provided with the basis for the decision. The obvious location for this information is in the first paragraph of Subsection 2.2.
2) The chemical form of many of the vitamins continues to be missing from the Supplementary Table. It is absolutely essential that this information be provided for each vitamin – without exception. What was the chemical form of vitamin A (palmitate? acetate?), vitamin E (numerous possibilities), vitamin K (K1? K2? K3?), thiamin, vitamin B6, biotin (d form?), pantothenic acid, niacin (actually included in the form of nicotinic acid?), vitamin B12?
3) Further to item 2, above, I note that S was included as a sulphate salt of K. Is the K added to the diet in sulphate form included in the total amount of K shown for the diet (3.6 g/kg)?
4) I requested, in relation to the original submission (item 1h of my Minor Concerns; Response 8 of the cover letter regarding Minor Concerns), that the basis be provided for the choice of zinc concentrations (15 and 30 mg/L) used in the in vitro experiment. This has not been done and, with apologies, the basis indicated in the cover letter (if I have understood correctly) is wholly inadequate. The first sentence of Response 8 in the cover letter reads “The zinc concentrations of 30 mg/L conducted in vitro was adopted from the mice treated with ZnLa LOW in the first experiment….”. I understand this to mean that a culture fluid zinc concentration of 30 mg/L was regarded as the equivalent, in vitro, of a dietary zinc level of 30 mg/kg (35.4 mg/kg as reported in line 110). With apologies, such an idea is entirely without foundation and is flawed at a fundamental level. No such relationship exists between dietary zinc levels and concentrations of zinc found in cell cultures.
Normally I would consider it essential to provide, explicitly in the manuscript, the basis for the culture fluid zinc levels used in the second experiment. However, if the dietary zinc levels of the first experiment actually were used as the basis, this was a huge and fundamental error in understanding of basic nutritional science, and no hint of this must be included in the manuscript. This said, the outcome of the experiment appears to be of some use in relation to the objective of the work – even if the basis for a key methodological detail (culture fluid zinc levels) was flawed.
Author Response

(The authors gave the same response as above.)
